

# ShallowBKGC: a BERT-enhanced shallow neural network model for knowledge graph completion

Ningning Jia and Cuiyou Yao

School of Management and Engineering, Capital University of Economics and Business, Beijing, China

## ABSTRACT

Knowledge graph completion aims to predict missing relations between entities in a knowledge graph. One of the effective ways for knowledge graph completion is knowledge graph embedding. However, existing embedding methods usually focus on developing deeper and more complex neural networks, or leveraging additional information, which inevitably increases computational complexity and is unfriendly to real-time applications. In this article, we propose an effective BERT-enhanced shallow neural network model for knowledge graph completion named ShallowBKGC. Specifically, given an entity pair, we first apply the pre-trained language model BERT to extract text features of head and tail entities. At the same time, we use the embedding layer to extract structure features of head and tail entities. Then the text and structure features are integrated into one entity-pair representation *via* average operation followed by a non-linear transformation. Finally, based on the entity-pair representation, we calculate probability of each relation through multi-label modeling to predict relations for the given entity pair. Experimental results on three benchmark datasets show that our model achieves a superior performance in comparison with baseline methods. The source code of this article can be obtained from https://github.com/Joni-gogogo/ShallowBKGC.

# INTRODUCTION

Knowledge graphs (KGs) such as DBpedia (*Auer et al., 2007*), Freebase (*Bollacker et al., 2008*), NELL (*Carlson et al., 2010*), and Wikidata (*Vrandečić & Krötzsch, 2014*) are important resources for many artificial intelligence tasks including semantic search (*Feddoul, 2020*), recommendations (*Wu et al., 2022*) and question answering (*Li & Moens, 2022*). These KGs are composed of factual triplets, with each triplet $(h, r, t)$ denotes the fact that relation $r$ exists between head entity $h$ and tail entity $t$. KGs can also be formalized as directed multi-relational graphs, where nodes correspond to entities and (labeled) edges represent the types of relationships among entities.

Although existing KGs usually contain more than billions of factual triplets, they still suffer from an incompleteness problem, *i.e.*, missing a large number of valid triplets (*Nguyen, 2020*). In particular, in English DBpedia 2014, 60% of person entities miss place-of-birth information, and 58% of the scientists have no facts about what they are known for

Corresponding author
Ningning Jia, jianingning@bupt.cn

*Krompaß, Baier & Tresp (2015)*. In Freebase, 71% of 3 million person entities miss place-of-birth information, 75% have no known nationality while 94% have no facts about their parents (*West et al., 2014*). Therefore, much efforts have focused on the knowledge graph completion (KGC) task, which aims to predict missing triplets in KGs by examining existing ones.

Knowledge graph embedding is a dominant approach for KGC, which maps entities and relations of a KG from a symbolic domain to a vector space and make predictions with their embeddings (*i.e.*, vectors). Among various knowledge graph embedding methods, deep neural network-based models such as convolutional neural network (*Dettmers et al., 2018*; *Dai Quoc Nguyen, Nguyen & Phung, 2018*), capsule network (*Nguyen et al., 2019*), graph neural network (*Schlichtkrull et al., 2018*; *Nguyen et al., 2022*; *Tong et al., 2023*), graph attention network (*Liang et al., 2023*; *Wang et al., 2023*) and generative adversarial network (*Li et al., 2023*) achieve state-of-the-art results.

In this article, we take a step back and propose a simple yet effective BERT-enhanced shallow neural network model for KGC, referred to as ShallowBKGC. Our motivation is based on the following considerations and observations: (1) deep neural network models are usually extended versions of simple shallow neural network models, and improving simple models can also produce corresponding improvements in complex deep models. (2) It has been demonstrated that neural networks with even one single hidden layer are universal approximators (*Demir, Moussallem & Ngomo, 2021*; *Ba & Caruana, 2014*), which means that shallow neural networks can learn almost any complex function previously learned by deep neural networks. Besides, the relatively low computational complexity of shallow neural networks makes them more suitable for large-scale KGs. Based on the above two observations, we prefer simple shallow neural network model instead of complex deep neural network model. It should be explained here that this work builds on a previous work (*Jia, 2022*), which we expand based on the next observation. *i.e.*, (3) Most neural network-based methods learn embeddings merely from structured triplets, ignoring rich text information contained in the entity name, which affects the accuracy of KGC. Recently, a pre-trained language model BERT (*Kenton & Toutanova, 2019*) has achieved great success on multiple natural language processing tasks. Some works (*Yao, Mao & Luo, 2019a*; *Kim et al., 2020*; *Zha, Chen & Yan, 2022*; *Wang et al., 2022*) represent entities and relations using their text information, and fine-tune BERT to infer the missing triplets. Although these works achieve appealing performance, they still fail to learn the structural information of a KG, and the fine-tune is computationally inefficient. In order to efficiently utilize both text and structural information, we apply BERT in the form of feature extraction to enhance the shallow neural network model to further improve the performance of KGC.

Our contributions in this article are summarized as follows:

- We propose ShallowBKGC, a BERT-enhanced shallow neural network model for KGC, which utilizes both text and structural information for this task.

- We introduce a pre-trained language model BERT in feature extraction manner to obtain text features of entities, thereby further improving the performance of KGC without retraining the proposed model.
- We conduct experiments on three benchmark datasets, and the experimental results demonstrate that our model achieves a superior performance in comparison with baseline methods.

## RELATED WORK

Existing KGC methods can be roughly classified into four categories: translation-based models, tensor decomposition-based models, neural network-based models and pre-trained language/large language-based models.

### Translation-based models

Translation-based models consider the relation between a head entity and a tail entity as a translation operation in the vector space and calculate the distance between the head entity vector and the tail entity vector to measure the plausibility of a triple. *Bordes et al. (2013)* present the initial translation-based model TransE, which learns low-dimensional and dense vectors for every entity and relation, so that relations correspond to translation vectors operating on vectors of entities. *Wang et al. (2014)* present TransH, which alleviates the complex relation problem in TransE by associating each relation with a relation-specific hyperplane. *Lin et al. (2015)* present a path-based TransE, named PTransE, which extends TransE by relation paths. *Nguyen et al. (2016)* present STransE that combines SE (*Bordes et al., 2011*) and TransE for KGC. *Sun et al. (2019)* present RotatE, which defines each relation as a rotation from head entity to tail entity in the complex space. *Le, Huynh & Le (2021)* present RotatH that combines RotatE and TransH for KGC.

Models of this category have the advantages of simplicity, intuitiveness, and high computational efficiency. However, research has shown that they have limitations in expressive power and are not suitable for non-Euclidean spaces.

### Tensor decomposition-based models

Tensor decomposition-based models use triangular norm to measure the plausibility of triplets. *Yang et al. (2015)* present DistMult, which considers triplets as tensor decomposition and constrains all relation embeddings to be diagonal matrices. ComplEx (*Trouillon et al., 2016*) extends DistMult to the complex space to better model asymmetric and inverse relations. *Balažević, Allen & Hospedales (2019)* present TuckER, which performs KGC based on tucker decomposition of binary tensors of known triplets. Inspired by the tucker decomposition of order-4 tensors, *Shao et al. (2022)* present a tensor decomposition model for temporal KGC. *Zhang et al. (2024)* extend tensor decomposition methods to temporal KGC.

Models of this category are proficient in capturing complex relations between entities and relations in KGs. However, as the scale of KGs grows, the computational complexity of these models may escalate rapidly.

## Neural network-based models

Various neural networks have been widely explored for KGC and achieved promising performance. *Dettmers et al. (2018)* present a multi-layer convolutional model ConvE, which explores convolutional neural network for KGC, and uses 2D convolution over embeddings to predict missing triplets in a KG. *Shang et al. (2019)* present an end-to-end graph structure-aware convolutional networks model SACN that combines graph convolutional network (GCN) and ConvE for KGC. *Dai Quoc Nguyen, Nguyen & Phung (2018)* present ConvKB, which utilizes convolutional neural network to capture the global relationships among dimensional entries of entity and relation embeddings. CapsE (*Nguyen et al., 2019*) combines ConvKB with capsule network for both KGC and search personalization tasks. *Schlichtkrull et al. (2018)* present relational graph convolutional networks and apply them to KGC. *Vashishth et al. (2020)* present CompGCN, which leverages a variety of composition operations from knowledge graph embedding techniques to jointly embed both entities and relation in a graph. SHALLOM (*Demir, Moussallem & Ngomo, 2021*) and the prior version of the model proposed in this article ASLEEP (*Jia, 2022*) apply shallow neural network for KGC and achieve good performance while maintaining high efficiency.

Models of this category have significant advantages in semantic feature learning, and our proposed model belongs to this category. The main difference between them and our model is that most of them usually rely on more deeper and complex neural networks while our model employs shallow neural network, which is not computationally demanding and friendly to real-time applications. Although there are several models that use shallow neural networks for KGC, these models only use structural information and ignore the rich information contained in text. Under the premise of keeping the model as simple as possible, we consider both text and structural information for KGC.

## Pre-trained language/large language-based models

Pre-trained language models and large language models have received widespread attention in many natural language processing tasks, including KGC. *Yao, Mao & Luo (2019a)* explore the pre-trained language model BERT for KGC. StAR (*Wang et al., 2021*) extends KG-BERT by taking into account structural information for KGC. *Yao et al. (2023)* present KG-LLM, which investigates large language models, including ChatGLM (*Du et al., 2022*) and LLaMA (*Touvron et al., 2023*) for KGC. *Yang, Fang & Zhou (2023)* present a constrained-prompt KGC based on large language model. *Zhang et al. (2023)* present KoPA, which integrates pre-trained KG structural features with large language model for KGC.

Models of this category achieve great success in KGC. However, these models usually require diverse fine-tuning strategies, and mostly cost much time in training and inference. It should be pointed out that the model we proposed also uses the pre-trained model BERT. The difference from the existing model is that in order to keep the model as simple as possible, we use BERT in a feature extraction manner, that is, the parameters in BERT are not involved in training.

## OUR PROPOSED MODEL

### Problem formulation

A KG is a type of multi-relational directed graph that typically consists of a collection of triplets in the form of $(h, r, t)$. It can be formally defined as $\mathcal{G} = (\mathcal{E}, \mathcal{R}, \mathcal{T})$, where $\mathcal{T}$ represents the set of all triplets, $\mathcal{E}$ and $\mathcal{R}$ represent the sets of all entities and relations respectively.

The objective of the KGC task is to predict missing relations in $\mathcal{G}$ based on the known triplets $\mathcal{T}$. In other words, the aim of KGC is to develop a model that accepts a query consisting of a head entity and a tail entity, $(h_i, ?, t_i)$, and ranks all candidate relations $r_c \in \mathcal{R}$ to resolve the query (*Lovelace & Rose, 2022*). An effective KGC model should enable correct candidates to have higher rankings than incorrect candidates.

### Model overview

Our proposed model ShallowBKGC takes as input an entity pair, and outputs the probability that each relation exists between the two entities. As illustrated in Fig. 1, our model consists of three key steps: (1) entity feature extraction, (2) entity-pair representation, and (3) multi-label relation modeling. The detailed calculation process of each step is as follows.

### Entity feature extraction

Given an entity pair $(h, t)$, our model extracts the features of the head and tail entities by taking into account both text and structural information.

For text information, we apply the pre-trained language model BERT (*Kenton & Toutanova, 2019*), which has achieved great success in multiple natural language processing tasks, to extract text feature of the given entity. Figure 2 illustrates the framework of BERT for entity text feature extraction. Formally, given the text information of head and tail entities, *i.e.*, head entity name $h\_text = \{w_1^h, w_2^h, ..., w_N^h\}$ and tail entity name $t\_text = \{w_1^t, w_2^t, ..., w_N^t\}$ (since many entities lack descriptive information and introducing additional information will increase computational complexity, we only use the name information that each entity has as text information), we first add a special classification token (CLS) and a separate token (SEP) at the beginning and end of the entity name respectively to obtain the marked entity name. Then through the tokenizer we obtain the representation of the marked entity name. Finally, we put the representation into BERT to get the text features of the head and tail entities as follows:

$$\mathbf{C} = \text{BERT}(\mathbf{CLS}, \mathbf{w}_1^h, \mathbf{w}_2^h, ..., \mathbf{w}_N^h, \mathbf{SEP}); \mathbf{h}_t = \mathbf{C} \tag{1}$$

$$\mathbf{C} = \text{BERT}(\mathbf{CLS}, \mathbf{w}_1^t, \mathbf{w}_2^t, ..., \mathbf{w}_N^t, \mathbf{SEP}); \mathbf{t}_t = \mathbf{C} \tag{2}$$

where $\mathbf{C} \in \mathbb{R}^H$ is the hidden vector of the special token (CLS), which contains the features of the entire input text. Therefore, we use it as the text feature of the given entity.

For structural information, our model receives the IDs of the given head and tail entities, and extracts the structure features of them through embedding layer as follows:

**Peer**J Computer Science

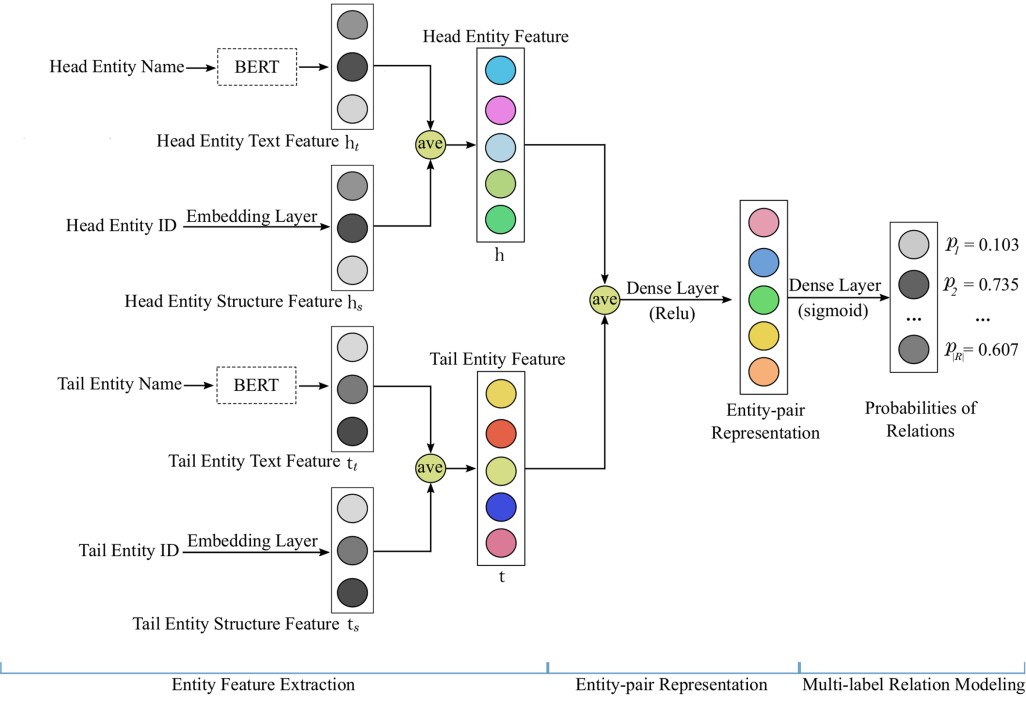

**Figure 1** **The architecture of our proposed model ShallowBKGC.**

$$\mathbf{h}_s = \text{Embedding\_layer}(\mathbf{h}_{\text{ID}}) \tag{3}$$
$$\mathbf{t}_s = \text{Embedding\_layer}(\mathbf{t}_{\text{ID}}) \tag{4}$$

where $\mathbf{h}_s \in \mathbb{R}^d$, $\mathbf{t}_s \in \mathbb{R}^d$ are embeddings of structural information corresponding to head and tail entities, respectively.

And then, we integrate entity text feature and entity structure feature through average operation, and get the entity feature as,

$$\mathbf{h} = \text{ave}(\mathbf{h}_t, \mathbf{h}_s) \tag{5}$$
$$\mathbf{t} = \text{ave}(\mathbf{t}_t, \mathbf{t}_s) \tag{6}$$

For the sake of computational convenience, and considering the consistency of tensor shapes, we intercept the first $d$ columns of text features when fusing text and structure features of entities.

## Entity-pair representation

After getting the features of entities, our model integrates head entity feature and tail entity feature into entity-pair representation through the average operation and a non-linear transformation as follows:

$$\mathbf{E} = \text{ReLU}(\mathbf{U} \cdot \text{ave}(\mathbf{h}, \mathbf{t})) \tag{7}$$

where $\mathbf{U} \in \mathbb{R}^{k \times k}$ is the transformation matrix, and ave() denotes the average function,

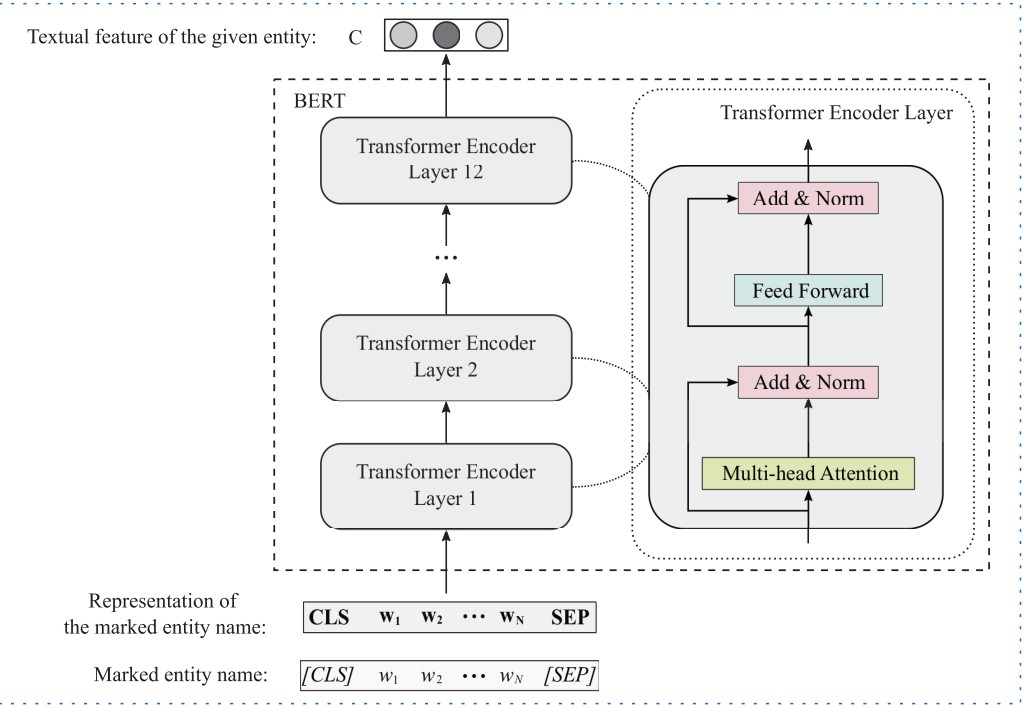

**Figure 2 The framework of BERT for entity text feature extraction.**

which aims to obtain the comprehensive features among each dimensional of head and tail entity features.

## Multi-label relation modeling

Since there may exist multiple relations between an entity pair, we model KGC as a multi-label learning problem. Based on the obtained entity-pair representation in the previous subsection, our model calculates the confidence scores for each relation as follows:

$$\mathbf{S} = \mathbf{V} \cdot \mathbf{E} \tag{8}$$

where $\mathbf{V} \in \mathbb{E}^{|R| \times k}$ is the collection of weight vectors for each relation. Afterwards, the sigmoid function is applied on each element of the score vector $\mathbf{S}$ to compute the probability of each relation to exist:

$$p_i = \frac{1}{1 + e^{-s_i}}, i = \{1, 2, ..., |R|\} \tag{9}$$

where $|R|$ denotes the number of relations.

## Model training

We define the loss function using cross-entropy as follows:

$$L = -\sum_i^{|R|} y_i \log(\mathrm{p_i}) + (1 - \mathrm{y_i}) \log(1 - \mathrm{p_i}) \tag{10}$$

where $y_i \in \{0, 1\}$ is the true value for relation $i$, $p_i$ is the predicted probability value for relation $i$. The loss function is optimized with Adam (*Kingma & Ba, 2015*), and dropout (*Srivastava et al., 2014*) is employed for regularization.

# EXPERIMENTS

## Datasets

We evaluate our model ShallowBKGC on three benchmark datasets: WN18RR (*Dettmers et al., 2018*), FB15k-237 (*Toutanova & Chen, 2015*) and YAGO3-10 (*Mahdisoltani, Biega & Suchanek, 2015*). WN18RR and FB15k-237 are derived from the lexical KG WordNet (*Miller, 1995*) and the real-world KG Freebase (*Bollacker et al., 2008*), respectively. YAGO3-10 is a dataset containing general facts from Wikipedia. The experimental datasets statistics are shown in Table 1.

## Evaluation metrics

We use mean rank (MR), mean reciprocal rank (MRR) and Hits@N as evaluation metrics, in which MR is the average rank of all test triplets, MRR is the average of the reciprocal ranks, and Hits@N is the percentage of test triplets that are ranked within top N. They are formally defined as follows:

$$\mathrm{MR} = \frac{1}{|Triplet_{test}|} \sum_{i=1}^{|Triplet_{test}|} rank_{triplet(i)} \tag{11}$$

$$\mathrm{MRR} = \frac{1}{|Triplet_{test}|} \sum_{i=1}^{|Triplet_{test}|} \frac{1}{rank_{triplet(i)}} \tag{12}$$

$$\mathrm{Hits@N} = \frac{|triplet(i) \in Triplet_{test} : rank_{triplet(i)} \leq N|}{|Triplet_{test}|} \tag{13}$$

where $|Triplet_{test}|$ is the number of test triplets, $triplet(i)$ is the $i$-th triplet.

Additionally, to evaluate the model efficiency, we measure the running time of the training phase. Record the average time of three epochs of the model on the dataset, in seconds. Our experimental platform is ModelArts, and the specific configuration selected is pytorch1.8-cuda10.2-cudnn7-ubuntu18.04, and a P100 GPU (16G).

## Baseline methods

We compare our model against the following state-of-the-art KGC models, including translation-based models TransE and RotatE, tensor decomposition-based models DistMult and ComplEx, neural network-based models ConvE, SHALLOM and ASLEEP, and pre-trained language/large language-based models KG-BERT, KG-ChatGLM-6B, KG-LLaMA-7B and KG-LLaMA-13B. Below we briefly introduce these models.

- **TransE** (*Bordes et al., 2013*) is the initial translation-based model that views relations as translations from head entities to tail entities on the low-dimensional space.
- **DistMult** (*Yang et al., 2015*) is a typical tensor decomposition-based model that restricts $n$-by-$n$ matrices representing relations to diagonal matrices.
- **ComplEx** (*Trouillon et al., 2016*) extends DistMult to the complex space.

**Table 1 Statistics of datasets.**

| Dataset | #Entities | #Relations | #Train | #Validation | #Test |
|---|---|---|---|---|---|
| FB15k-237 | 14,541 | 237 | 272,115 | 17,535 | 20,466 |
| WN18RR | 40,943 | 11 | 86,835 | 3,034 | 3,134 |
| YAGO3-10 | 123,182 | 37 | 1,079,040 | 5,000 | 5,000 |

Note:
#Entities denotes the number of all unique entities. #Relations denotes the number of all unique relations. #Train, #Validation and #Test denote the number of triplets contained in train set, validation set and test set, respectively.

- **RotatE** (*Sun et al., 2019*) is an efficient ranslation-based model that represents entities as complex vectors and relations as rotations.
- **ConvE** (*Dettmers et al., 2018*) is a deep neural network-based model that applies convolutional neural network for KGC.
- **SHALLOM** (*Demir, Moussallem & Ngomo, 2021*) is a shallow neural network-based model for KGC.
- **ASLEEP** (*Jia, 2022*) improves the way SHALLOM obtains entity pair representation, and is the prior version of our proposed model.
- **KG-BERT** (*Yao, Mao & Luo, 2019b*) is a pre-trained language-based model that firstly employs BERT to KGC.
- **KG-ChatGLM-6B, KG-LLaMA-7B, KG-LLaMA-13B** (*Yao et al., 2023*) are large language-based models that perform instruction tuning with ChatGLM (*Du et al., 2022*) and LLaMA (*Touvron et al., 2023*) for KGC.

## Hyperparameter optimization

We select the hyperparameters of ShallowBKGC by grid search based on Hits@1 of the relation prediction task on the validation set of each dataset. We manually specify the hyperparameter ranges: embedding size among $\{50, 100, 150\}$, epochs among $\{50, 100, 150\}$, batch size among $\{512, 1000\}$, dropout rate among $\{0.25, 0.5\}$, and $L_2$-normalizer among $\{0.1, 0.01, 0.001\}$. Table 2 shows parameters values in the experiments.

## Experimental results

In this section, we compare the performance of our model ShallowBKGC with that of the baseline methods on the widely used relation prediction task. The task of relation prediction is to complete a triplet $(h, r, t)$ with $r$ missing, *i.e.*, to predict the missing $r$ given $(h, t)$. From the relation prediction results shown in Tables 3–5, we summarize our key observations in the following section.

(1) The shallow neural network-based models, *i.e.*, our model ShallowBKGC and the baselines ASLEEP and SHALLOM, outperform the translation-based model TransE, the complex vector-based model ComplEx and RotatE, the deep neural network-based model ConvE, the pre-trained language-based model KG-BERT, and even the large language-based model KG-ChatGLM-6B, demonstrating the effectiveness of shallow neural network for KGC. For example, compared to TransE, DistMult, ComplEx and RotatE, our model

**Table 2  Hyperparameter values.**

| Dataset | FB15k-237 | WN18RR | YAGO3-10 |
|---|---|---|---|
| Embedding size | 50 | 100 | 50 |
| Epochs | 150 | 100 | 100 |
| Batch size | 1,000 | 512 | 512 |
| $L_2$-normalizer | 0.1 | 0.1 | 0.1 |

**Table 3  Relation prediction results on FB15k-237.**

| Model | MR | MRR | Hits@1 | Hits@3 | RT |
|---|---|---|---|---|---|
| TransE (*Bordes et al., 2013*)* | 1.352 | 0.966 | 0.946 | 0.984 | 31 s |
| DistMult (*Yang et al., 2015*)* | 1.927 | 0.875 | 0.806 | 0.936 | 43 s |
| ComplEx (*Trouillon et al., 2016*)* | 1.494 | 0.924 | 0.879 | 0.970 | 112 s |
| RotatE (*Sun et al., 2019*)* | 1.315 | 0.970 | 0.951 | 0.980 | >600 s |
| ConvE (*Dettmers et al., 2018*)* | – | 0.667 | 0.562 | 0.732 | 33 s |
| SHALLOM (*Demir, Moussallem & Ngomo, 2021*)[†] | **1.106** | 0.969 | 0.947 | 0.992 | 2 s |
| ASLEEP (*Jia, 2022*) | 1.109 | 0.970 | 0.949 | 0.992 | 2 s |
| ShallowBKGC | 1.108 | **0.972** | **0.952** | **0.993** | 3 s |

Note:
The best score is in bold, while the second best score is underlined. Results marked * are taken from *Wang, Ren & Leskovec (2020)* and *Demir & Ngomo (2021)*, respectively. † denotes results from our re-implementation. RT is the abbreviation for running time.

**Table 4  Relation prediction results on WN18RR.**

| Model | MR | MRR | Hits@1 | Hits@3 | RT |
|---|---|---|---|---|---|
| TransE (*Bordes et al., 2013*)* | 2.079 | 0.784 | 0.669 | 0.870 | 36 s |
| DistMult (*Yang et al., 2015*)* | 2.024 | 0.847 | 0.787 | 0.891 | 16 s |
| ComplEx (*Trouillon et al., 2016*)* | 2.053 | 0.840 | 0.777 | 0.880 | 38 s |
| RotatE (*Sun et al., 2019*)* | 2.284 | 0.799 | 0.735 | 0.823 | >600 s |
| ConvE (*Dettmers et al., 2018*)* | – | 0.353 | 0.143 | 0.405 | 35 s |
| SHALLOM (*Demir, Moussallem & Ngomo, 2021*)[†] | 1.201 | 0.925 | 0.866 | 0.985 | 3 s |
| ASLEEP (*Jia, 2022*) | 1.176 | 0.934 | 0.883 | 0.985 | 3 s |
| ShallowBKGC | **1.125** | **0.949** | **0.908** | **0.992** | 4 s |

Note:
The best score is in bold, while the second best score is underlined. Results marked * are taken from *Wang, Ren & Leskovec (2020)* and *Demir & Ngomo (2021)*, respectively. y denotes results from our re-implementation. RT is the abbreviation for running time.

ShallowBKGC achieves 45.8%, 44.4%, 45.2%, and 50.7% relative improvements in MR on WN18RR, respectively. Compared to ConvE, our model ShallowBKGC achieves 76.5% and 38.9% absolute improvements in Hits@1 on WN18RR and FB15k-237, respectively. Compared to KG-BERT and KG-ChatGLM-6B, our model ShallowBKGC achieves 0.2% and 11.7% absolute improvements in Hits@1 on YAGO3-10, respectively. It is worth mentioning that the result of KG-ChatGLM-6B is lower than that of KG-BERT. This

**Table 5  Relation prediction results on YAGO3-10.**

| Model | MR | MRR | Hits@1 | Hits@3 | RT |
|---|---|---|---|---|---|
| KG-BERT (*Yao, Mao & Luo, 2019a*)* | – | – | 0.681 | – | >1,300 s |
| KG-ChatGLM-6B (*Yao et al., 2023*)* | – | – | 0.566 | – | – |
| KG-LLaMA-7B (*Yao et al., 2023*)* | – | – | **0.702** | – | – |
| KG-LLaMA-13B (*Yao et al., 2023*)* | – | – | 0.696 | – | – |
| SHALLOM (*Demir, Moussallem & Ngomo, 2021*)[†] | 1.465 | 0.776 | 0.556 | 0.994 | 47 s |
| ASLEEP (*Jia, 2022*)[†] | 1.388 | 0.813 | 0.630 | 0.996 | 47 s |
| ShallowBKGC | **1.301** | **0.837** | 0.683 | **0.996** | 51 s |

**Note:**
Results marked [*] are taken from *Yao et al. (2023)*. The dash (–) denotes values missing. The best score is in bold, while the second best score is underlined.

suggests that it is not the case that the more layers a model has or the newer the technology is, the better the results will be.

(2) Comparing our model ShallowBKGC with ASLEEP and SHALLOM, we can see that the MRR, Hits@1 and Hits@3 values of ShallowBKGC on the three datasets are better than ASLEEP and SHALLOM. This indicates that it is beneficial to take both text and structural information into account for KGC. Because the main difference between our model ShallowBKGC and the baselines ASLEEP and SHALLOM is that our model ShallowBKGC combines text and structural information for KGC, while ASLEEP and SHALLOM merely rely on structural information.

(3) The RTs of our model ShallowBKGC on three benchmark datasets significantly outperform the baselines, which shows the efficiency of our model. It should be noted that in order to minimize the impact of programming differences, we use OpenKE (*Han et al., 2018*) to reproduce the running times of TransE, DistMult, ComplEx and RotatE. The RTs of ConvE, SHALLOM, ASLEEP and KG-BERT are obtained from their corresponding source codes. Due to permission issues, the RTs of KG-ChatGLM-6B, KG-LLaMA-7B and KG-LLaMA-13B are missing. However, from the perspective of the number of layers and parameter scale, the RTs of these models are likely to be larger than KG-BERT. More formally, the time complexity of our model is $O(d_e)$ (where $d_e$ represents the dimension of entities), which is the same as that of the baselines, except the language-based baselines. From *Wang et al. (2021)*, we can see that the most relevant language-based baseline KG-BERT's time complexity is $O(L_t^2 |E|^2 |R|)$, where $L_t$ is the length of triple text, $|E|$ and $|R|$ are the numbers of entities and relations respectively. Additionally, the space complexity of our model is $O(L_e |E| d_{token} + |E| d_e)$, where $L_e$ is the length of entity text, $d_{token}$ is the dimension of entity text tokens.

(4) It is worth noting that the results of ConvE are significantly lower than those of the other models, probably because it relies on an improper pre-trained model for initialization, and is trained on entity prediction task (*i.e.*, given $(h, r)$ predict $t$, or given $(r, t)$ predict $h$) but tested on relation prediction task. It has been demonstrated that the initialization, hyperparameter optimization, and training strategies have significant effects on prediction performance (*Demir & Ngomo, 2021*; *Ruffinelli, Broscheit & Gemulla, 2020*).

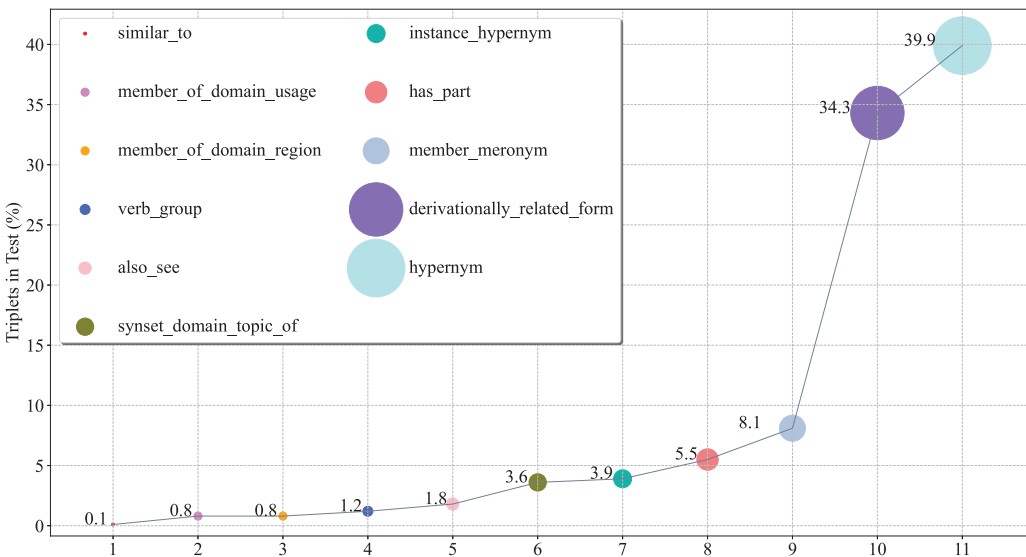

**Figure 3 The percentage of triplets corresponding to relations on the WN18RR test set.**

In contrast, our model ShallowBKGC is as simple as possible, it does not require retraining the pre-trained model BERT, special hyperparameter optimization approach, or complex training strategy, thus minimizing model uncertainty.

## Fine-grained performance analysis

To further verify the capacity of our model from a fine-grained perspective, we plot the percentage of each relation on WN18RR in Fig. 3, and report the Hits@N and MRR performance on each relation in Figs. 4 and 5, respectively.

From these figures, we can observe that:

(1) From the perspective of Hits@N, there are five relations, *i.e.*, _similar_to, _member_of_domain_region, _instance_hypernym, _member_meronym, and _derivationally_related_form, Hits@1 values exceed 90%. Ten relations Hits@3 values exceed 90%. In particular, there are four relations, *i.e.*, _similar_to, _member_of_domain_region, _verb_group and _hypernym, Hits@3 values reached 100%. Thus, the results are consistent with Table 4, which further demonstrates the effective of our model at a fine-grained level.

(2) There are eight relations with MRR values exceeding 90%. They are _similar_to, _member_of_domain_usage, _member_of_domain_region, _synset_domain_topic_of, _instance_hypernym, _member_meronym, _derivationally_related_form and _hypernym. Moreover, five of these eight relations Hits@1 values exceed 90%. This experimental result once again demonstrates the effectiveness of our model and the consistency of the results at a fine-grained level.

## Ablation study

We conduct ablation studies to provide a more detailed analysis of the effectiveness of each part of our model. The models used for comparison are the following ones: (a)

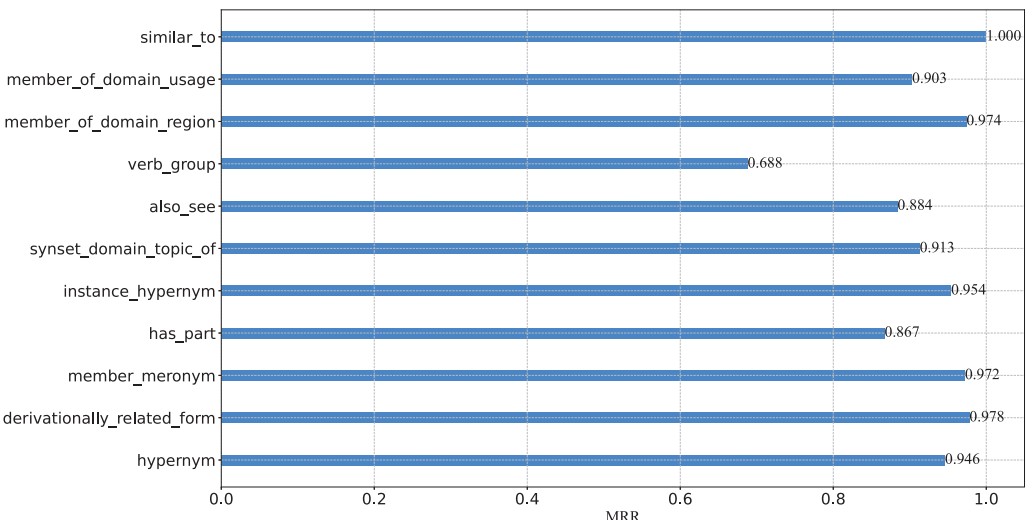

**Figure 4** Hits@N results on the WN18RR test set.

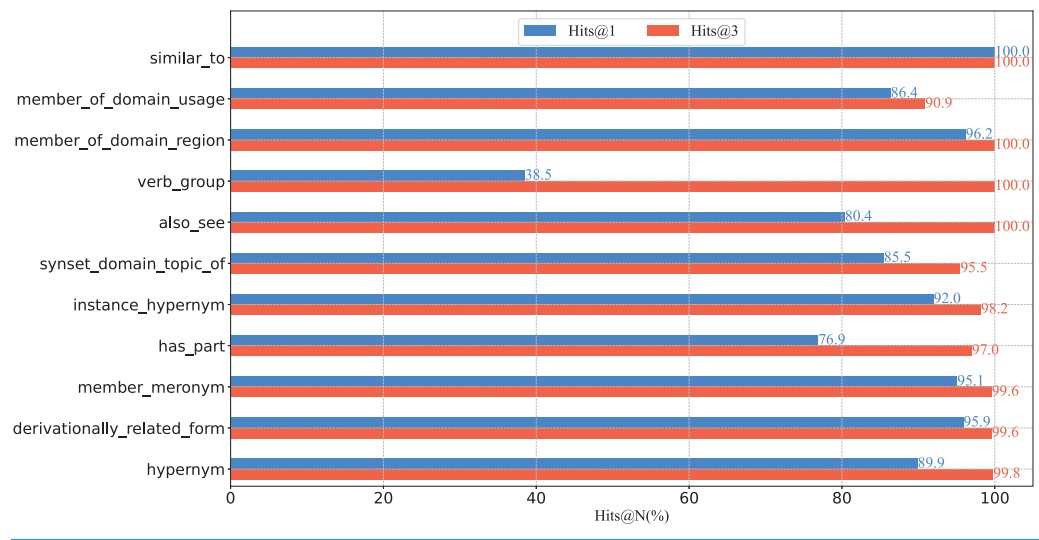

**Figure 5** MRR results on the WN18RR test set.

ShallowBKGC-Text is the model that merely rely on text information, and (b) ShallowBKGC-Structure is the model that merely rely on structural information.

Table 6 shows the relation prediction results on three datasets, *i.e.*, FB15k-237, WN18RR, and YAGO3-10. From which we can see that:

(1) ShallowBKGC outperforms ShallowBKGC-Text and ShallowBKGC-Structure in all three datasets, indicating that considering both text and structural information is beneficial to KGC. This is also consistent with the results of the previous experiments.

(2) ShallowBKGC-Structure achieves better results than ShallowBKGC-Text, the main reason is that we set the entity text feature obtained by BERT untrainable to reduce the computational complexity, this sacrifices the performance to a certain extent. It is worth further explaining that from Tables 5 and 6, we can see that ShallowBKGC-Text

**Table 6 Ablation study tests on the three datasets.**

| Dataset | Model | MR | MRR | Hits@1 | Hits@3 | RT |
|---|---|---|---|---|---|---|
| FB15k-237 | ShallowBKGC-Text | 1.537 | 0.908 | 0.859 | 0.950 | 1 s |
| | ShallowBKGC-Structure | 1.153 | 0.963 | 0.938 | 0.987 | 2 s |
| | ShallowBKGC | **1.108** | **0.972** | **0.952** | **0.993** | 3 s |
| WN18RR | ShallowBKGC-Text | 1.307 | 0.891 | 0.813 | 0.971 | 1 s |
| | ShallowBKGC-Structure | 1.163 | 0.932 | 0.876 | 0.991 | 3 s |
| | ShallowBKGC | **1.125** | **0.949** | **0.908** | **0.992** | 4 s |
| YAGO3-10 | ShallowBKGC-Text | 1.471 | 0.783 | 0.559 | 0.994 | 35 s |
| | ShallowBKGC-Structure | 1.375 | 0.819 | 0.637 | 0.994 | 47 s |
| | ShallowBKGC | **1.301** | **0.837** | **0.683** | **0.996** | 51 s |

Note:
The best score of each dataset is in bold.

outperforms KG-ChatGLM-6B, considering that the latter has far more parameters than the former, but the experimental results are very close, which also reflects the efficiency of our model.

(3) ShallowBKGC-Text has the shortest RT because it has the fewest parameters. The RTs of ShallowBKGC-structure and ShallowBKGC are close, indicating that our model do not spend much time fusing text and structural information.

## CONCLUSION

In this article, we propose a simple yet effective BERT-enhanced shallow neural network model for KGC that jointly considers text and structural information. Specifically, given an entity pair and the text information of the entities, our model first extracts the text features of the entities by BERT in a feature extraction manner, and extracts the structure features of the entities through the embedding layer. Then the text and structure features of the head and tail entities are integrated into an entity-pair representation through an average operation and a non-linear transformation, which aims to obtain the comprehensive and rich features of the entities. Finally, based on the entity-pair representation and considering that multiple relations may exist between entities, our model calculates the probability of each relation through multi-label modeling. Experimental results on three public datasets shown that our model achieves a superior performance in comparison with the baseline methods.

In the future, we plan to (1) further study the performance of our model on two KGC related tasks, *i.e.*, triplet classification and entity prediction; (2) extend our model to temporal KGC and link prediction in social networks tasks; (3) explore the possibilities and performance of shallow neural networks on other tasks that can be organized into triplets.

## ACKNOWLEDGEMENTS

We thank the anonymous reviewers for their insightful comments and constructive suggestions.

### Funding

This work was supported by the Capital University of Economics and Business under grant 01892254413027. The funders had no role in study design, data collection and analysis, decision to publish, or preparation of the manuscript.

### Grant Disclosures

The following grant information was disclosed by the authors:
Capital University of Economics and Business: 01892254413027.

### Competing Interests

The authors declare that they have no competing interests.

### Author Contributions

- Ningning Jia conceived and designed the experiments, analyzed the data, performed the computation work, prepared figures and/or tables, and approved the final draft.
- Cuiyou Yao performed the experiments, authored or reviewed drafts of the article, and approved the final draft.

### Data Availability

The code is available at GitHub and Zenodo:
- https://github.com/Joni-gogogo/ShallowBKGC/tree/1.1.
- Joni-gogogo. (2024). Joni-gogogo/ShallowBKGC: knowledge graph completion (1.1). Zenodo. https://doi.org/10.5281/zenodo.10791673.

### Supplemental Information

Supplemental information for this article can be found online at http://dx.doi.org/10.7717/peerj-cs.2058#supplemental-information.

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
