# Peer review of "ShallowBKGC: a BERT-enhanced shallow neural network model for knowledge graph completion"

_PeerJ Computer Science, doi:10.7717/peerj-cs.2058_

## Round 0.1 · original submission · Major Revisions

Dear authors.

Based on the reports submitted by the three reviewers, I made a decision of "major revision".

I think your research is very useful, but, you should work on the improvements suggested by the reviewers.

·

Basic reporting

This paper presents a shallow neural network model for knowledge graph completion, that using text and structural information of head and tail entities, is able to rank possible relations based on the trained ones. Its evaluation demonstrates an improvement in comparison with baseline methods. The paper is well presented and structured and the idea is exposed clearly, moreover, the introduction really helps to introduce the problem even for someone that is not really versed in the topic, which makes the whole paper self-contained.

However there are some aspects that need some revision because without correcting them it is impossible to totally evaluate the overall solution and therefore advice in favour of its publication.

First of all, it is mentioned that a prior version of this paper has already been published. I would suggest that the authors reference this paper more explicitly via a bibliographic citation and explain in more detail the differences of the current version with the previous one. It is also interesting that through the evaluation one can guess that the prior version is the ASLEEP model so it is a bit strange the decision of the authors of not making this more explicit.

Experimental design

While the proposed model is easy to follow in the theoretical explanation, when taking a look at the source code on Github this becomes less evident. There is no README provided (even though a very succinct RreadMe.txt exists in the repository) which makes it very hard to reproduce the authors findings, reproduce the evaluation, or understand in more detail the proposed model. I would advise to put some effort in the documentation of the repository and also restructure the code a little bit to make it more readable and understandable.

Validity of the findings

Then, the related work reads more like a list of works rather than a proper revision of different state-of-the-art methods, highlighting the differences between them and contrasting them (with the advantages and disadvantages) with the proposed solution.

Similarly the results reporting is quite messy and it would benefit from making a list out of it while using the text to highlight the observed differences and/or to establish a bit of discussion on why this is happening. Also, Figures 2 and 3 seem to contain interesting data, but it is impossible to read them in a printed version.

Additional comments

More comments and typos per section
legend (#line) extract from the paper → rephrased extract [comment]
#Introduction
(#67) We introduce pre-trained → We introduce a pre-trained
#Our proposed model
(#125) of a triplet do not exist → of a triplet that does not exist
(#160) After get the features of entities → After getting the features of entities
(#170) the probability of each relation exists → the probability of each relation to exist
#Experiments
(#180) the refined version (eliminate → the refined version (which eliminates
(#215) Tables 4,3 → Tables 3 and 4
#Table 3
ASLEEP Jia (2022) → ASLEEP (Jia, 2022)

·

Basic reporting

This paper proposes a simple and efficient method called ShallowBKGC for knowledge graph completion. Specifically, it adopts the pre-trained BERT to capture the entities' textural features. These features are fused with KG embedding to inject the structure infroamtion. Then, two simple dense layers are adopted for predictions. Experiment results on several datasets demonstrate the effectveness of proposed method.

Experimental design

Experiment are conducted on two widely used benchmark datasets. Results show that proposed method outperforms some baselines.

However, there are comments to the experiments.

1. The proposed method utilize the text information. However, all the used baselines only use the structure information. Other text-based KGC methods, e,g., KG-BERT, KGLLM, StAR should be compared.

2. Since the proposed method claims to be a efficient method, complutation time should be compared.

3. Ablation studies are required to show the effectiveness of text and structure embeddings.

Validity of the findings

This paper proposes a simple shallow model for knowledge graph completion. However, the experiment results cannot fully support the motivation right now.

1. The proposed method still need the BERT in the forward stage, which could introduce high complutation cost.

2. Whether the parameters of BERT requiring tuning during training are not discussed.

3. Compared baseslines are out-of-dated. Some relative methods are not complared.

4. Lacks of ablation studies.

Additional comments

The text in Figures 2 and 3 is too small to be legible.

·

Basic reporting

General Comments
Summary
This paper seeks to tackle the problem of knowledge graph completion by proposing ShallowBKGC. ShallowBKGC is a BERT-enhanced shallow neural network model capable of solving the problem of knowledge graph completion. The authors use this pre-trained language model BERT to extract the features of a pair of head and tail entities, and then it calculate the probability that there exists a relationship between them.

This paper is well-written. It presents a well-supported description of the proposed solution with a visual representation of ShallowBKGC and all the supporting algebra that defines how the model is trained. In the experiments, the proposed solution is compared against multiple state-of-the-art techniques to determine its performance stacks against the state-of-the-art. For this purpose, two benchmarks are used. These experiments illustrate that ShallowBKGC can outperform state-of-the-art techniques.

This paper presents a couple of negative points that must be addressed. The first is that there are words that are misspelled and phrases that do not make sense, for example, in the Related Work section, you can find the word “presnet” it should be “presents,” and in the Our Proposed Model section, you can find the sentence “correctness of a triple do not exists in the given knowledge graph” it should be “correctness of a triple that does not exist in the given knowledge graph.” The placing of some figures (in particular Figure 2) and tables (in particular Tables 3 and 4) breaks the flow of the text, causing the reader to lose their train of thought. In the analysis of the experimental results, the inclusion of mathematical operations in the text seems to be out of place, and given how the sentences are structured, it would seem that only the result of the operations is needed. Finally, the statement of the problem is not well formalized.

This paper presents a very good approach to the problem of knowledge graph completeness with an extensive experimental study. But, before publishing it, the authors must address the formalization of the problem, the placing of figures and tables, and grammar errors.

Section Comments
Section 1: Introduction
Positive: This section does a good job of informing the reader of what to expect from this paper. It presents background knowledge to understand some concepts to understand the proposed solution, as well as the motivation behind this work. Additionally, list the contribution of this work.
Negative:

Section 2: Related Work
Positive: This section introduces multiple previous works and techniques that use knowledge embeddings to solve the problem of knowledge graph completion. Some of these methods were then used in the experimental study.
Negative: This section presents some tupos and phrases that are misswritten. For example, you can find the word “presnet” should be “presents,” and “structural information of knowledge graph” should be “structural information of a knowledge graph”. It would be good for this section to describe how the proposed solution differs from all these previous works.

Section 3: Our Proposed Model
Positive: This section describes the problem and the proposed solution ShallowBKGC. It gives an overview of ShallowBKGC and a figure that illustrates the inner workings of the technique. Each layer of the neural network and how the model is trained are described in this section.
Negative: The problem is not well formalized. The author mentioned that the problem is formalized in a different paper; just mentioned this does not actualize the problem. Therefore, the authors should formalize the problem properly and reference the corresponding paper. Multiple grammar errors can be found in this section, like “correctness of a triple do not exists in the given knowledge graph”, it should be “correctness of a triple that does not exist in the given knowledge graph,” and “After get the features of entities” should be “After getting the features of the entities.” It would be recommended to center the equations presented in this section; this is just a suggestion and not a demand.

Section 4: Experiments
Positive: This section describes the experimental configuration and results. It illustrates that two datasets are used, WN18RR and FB15k-237, and seven state-of-the-art techniques besides the proposed solution. The performance of each technique is analyzed in detail, and the main conclusion is that ShallowBKGC outperforms the other techniques. This section also presents multiple tables and figures that help to visualize the results of the experiments.
Negative: The main issue of this section is that given the location of Tables 3 and 4, they break the flow of the text and cause confusion for the reader. This also happens with Figure 2 but to a lesser extent. Additionally, there is a lot of empty space between these tables. The reference to Tables 3 and 4 in the text is confusing; please replace “Tables 4,3” with “Tables 3 and 4.” Including how the mathematical operation of the relative improvement is calculated in the text seems out of place, especially given how the sentences are structured since they suggest that they only need the result of the operation, not the entire operation. I would suggest moving the computation of the relative improvement to either the existing tables or creating a table for them. I suggest moving Figure 2 to the top of the page. It would be recommended to center the equations describing the metrics presented in this section; this is just a suggestion and not a demand.

Section 5: Conclusion
Positive: This section summarizes the proposed solution well and mentions how it outperforms the state-of-the-art techniques.
Negative: Adding an additional sentence or two to describe future directions for this work would be recommended.

Experimental design

The authors define a quite robust experimental study. They used two well-known datasets and even state-of-the-art techniques besides the proposed solution. They used three metrics and gave the equations for their calculation. All the results are presented in tables or figures, highlighting the results that presented the best performance. The proposed solution, the datasets, the other state-of-the-art techniques, and the metrics are publicly available. Therefore, all the experiments can be reproduced. Unfortunately, the authors did not define any research questions.

Validity of the findings

Given the results, the authors have proposed a new method for knowledge graph completion that outperforms other state-of-the-art techniques. The conclusions reflect the results achieved from the experiments well. Therefore, this paper presents an innovative contribution to the area.

---

## Round 0.2 · Minor Revisions

Please, pay attention to the reports of the reviewers.

·

Basic reporting

This paper is a resubmission of an already reviewed paper that received a major revision. From the last version of it a substantial improvement can be appreciated, so first of all, I would like to thank the authors for it. This new version solves many issues that were present in the previous one while it adds some extra experiments and information which make the paper even more interesting. Before I can recommend its publication there are some minor issues that need to be resolved.

Most of my comments are related to grammatical and orthographic errors (which the authors can find at the bottom of my review) upon which I highly encourage the authors to take good care of this aspect and perform an in-depth proofreading before submitting the final version.

Experimental design

I really appreciate the effort put on the Github repository and I would only make an additional recommendation which is creating an archived version on Zenodo. This will provide a DOI for your code, it will make it more findable and discoverable by other researchers and it will help to identify better the version of the code that corresponds to this paper. This is very much inline with the recommendations that this journal gives to authors: https://peerj.com/about/author-instructions/cs

Validity of the findings

The conclusions are still a bit poor as right now this section is merely a very short summary of the paper without any further details nor any special information. It is very similar to the abstract but with even less details. The future work part is only one sentence which does not give much hints on how the authors plan to evolve this work.

Additional comments

More comments and typos per section
legend (#line) extract from the paper → rephrased extract [comment]

#Abstract
(#21) model achieves superior performance → [in comparison with?]

#Introduction
(#32) suffer from incompleteness problem → suffer from an incompleteness problem
(#39) relations of knowledge graph → relations of knowledge graphs
* The preliminary […] on neural information processing → [I guess this footnote can removed]
(#53-54) Therefore, we prefer simple […] → [why?]
(#54-55) Portions of this text were previously published as part of a conference paper […] → This work builds on a previous work expanding it by …
(#63) information of knowledge graph → information of a knowledge graph
(#65) knowledge graph → KG [it is possible to introduce this acronym earlier and then use it throughout the paper]
(#74) model achieves superior performance → model achieves superior performance in comparison with …

#Related work
(#126) several models use shallow → several models that use shallow
#Our proposed model
(#150) The objective of knowledge graph completion task → The objective of the knowledge graph completion task
(#150) missing relation → missing relations

#Experiments
(#209) very competitive → [What do you mean by very competitive?]
(#Table 1) #Entity → #Entities
(#Table 1) #Relation→ #Relations
(#Table 1) #Valid → #Validation [?]
(#Table 1) [Units are missing so it is difficult to know of what exactly we are talking about]
(#221) P100 GPU (16G) → [This is somewhat cryptic, more information is needed, including also more information on the hardware than just the GPU]
(#235) model that applying → model that applies
(#Tables 3, 4 and 5) is in underline → is underlined
(#268) or the newer the technology → or the newer the technology is
(#282) the RTs of these models is likely → the RTs of these models are likely
(#285) but test on relation → but tested on relation
(#Figure 3) [If possible, increase the size a little bit]
(#296-297) [Rephrase, right now the sentence is not understandable]
(#Figures 4 and 5) w.r.t → w.r.t.
(#308) are the followings → are the following ones
(Table 6) Ablation on three datasets → Ablation study tests on the three datasets

#Conclusion
(#328) model extract the text → model extracts the text
(#328) BERT in feature extraction manner → BERT in a feature extraction manner
(#328-329) and extract the structure features → and extracts the structure features

·

Basic reporting

I appreciate the author's response. Most of my comments have been addressed.

Experimental design

1. Instead of the running time, I am also curious about the memory consumption of the proposed method compared to the baselines due to the introduction of Bert during inference.

Validity of the findings

no comment

Additional comments

no comment

·

Basic reporting

General Comments
This paper seeks to tackle the problem of knowledge graph completion by proposing ShallowBKGC. ShallowBKGC is a BERT-enhanced shallow neural network model capable of solving this problem. The authors use this pre-trained language model BERT to extract the features of a pair of head and tail entities and calculate the probability that a relationship exists between them.

The paper is well-written. The authors considered all the comments from the previous version and presented a much better version. The paper is accepted. Before publishing, the authors should fix some small mistakes. These mistakes are described below.

Recommendations
In the Abstract, the phrase “more deeper and complex” should be “deeper and more complex.”
In the Abstract, the phrase “our model archives superior performance” is missing an “a” before “superior performance”; it should be “our model archives a superior performance.”
In the Introduction, replace “recommendation” with “recommendations.”
In the Introduction, the phrase “missing a lot of valid triplets” is too informal. It is more formal to replace “a lot” with “multiple,” “large number,” or “various.”
In the first contribution in the introduction, the phrase “for knowledge graph completion” is used twice in the same section. Replace the second use with “for this purpose” or “for this task.”
In the experiments section, the header of Table 1 does not clearly state what “#Train” is.
The symbols for Tables 3 and 4 are described similarly in the experiments section. It would be beneficial to explain the symbols once in the description of Table 3 and then say that they are used in the same manner in Table 4.
In the descriptions for Tables 3, 4, and 5, the phrase “is in underline” should be “is underlined.”
In the experimental results section, the phrase “we summarize key observations as below” should be “we summarize our key observations in the following section.”

Experimental design

Three datasets, WN18RR, FB15k-237, and YAGO3-10, are used. Additionally, 11 state-of-the-art techniques, in addition to the proposed solution, are used. The performance of each technique is analyzed in detail, and the main conclusion is that ShallowBKGC outperforms the other techniques. The authors presented multiple tables and figures that helped visualize the results of the experiments. This version of the experimental configuration used is much more extended than the previous version of the paper.

Validity of the findings

The authors' proposed solution outperforms current state-of-the-art approaches in three different datasets. The results are analyzed in detail, describing the performance of all the approaches used in the experiments. This can, in turn, inform other authors of their behavior for future works.

---

## Round 0.3 · accepted · Accept

Congratulations!

After reviewing the reports of the 3 reviewers, it is clear that the manuscript has improved enough to be accepted.

·

Basic reporting

This paper is a revision of an already submitted and revised paper. From the two revision iterations the paper has improved a lot and all my remarks and comments have been addressed. Therefore, I can advise in favour of its publication. I also want to congratulate the authors for their work and the great outcome that this paper supposes.

Experimental design

no comment

Validity of the findings

no comment

Additional comments

Some minor typos per section
legend (#line) extract from the paper → rephrased extract [comment]

#Introduction
(#34) missing large number of valid triplets → missing a large number of valid triplets
(#62-63) contained in entity name → contained in the entity name

#Experiments
(#319) with MRR values exceed 90% → with MRR values exceeding 90%

·

Basic reporting

All of my concerns have been addressed. I am in champion of accepting this manuscript with the current version.

Experimental design

N/A

Validity of the findings

N/A

·

Basic reporting

This iteration of the paper is very well written, and I have no further comments. The proposed solution presents good performance and can be used as a point comparison for possible future works.

Experimental design

The experimental configuration presented in this work is quite extensive. The authors used three well-known datasets, multiple state-of-the-art techniques, and multiple metrics alongside the proposed solution to determine how the proposed solution compares against state-of-the-art techniques. From the results, it could be seen that the proposed solution outperforms state-of-the-art techniques.

Validity of the findings

Given how extensive the experimental study was and the results obtained, it could be concluded that the work presented in this paper is a good contribution to the literature.